# Patients with Hematological Malignancies Treated with T-Cell or B-Cell Immunotherapy Remain at High Risk of Severe Forms of COVID-19 in the Omicron Era

**DOI:** 10.3390/v14112377

**Published:** 2022-10-27

**Authors:** Jeremie Zerbit, Marion Detroit, Antoine Meyer, Justine Decroocq, Benedicte Deau-Fischer, Paul Deschamps, Rudy Birsen, Johanna Mondesir, Patricia Franchi, Elsa Miekoutima, Corinne Guerin, Rui Batista, Didier Bouscary, Lise Willems, Marguerite Vignon

**Affiliations:** 1Department of Pharmacy, Hospital at Home, University Hospitals of Paris, 75014 Paris, France; 2Clinical Pharmacy, Cochin Hospital, University Hospitals of Paris, 75014 Paris, France; 3Clinical Hematology, Cochin Hospital, University Hospitals of Paris, 75014 Paris, France; 4Institut Cochin, CNRS UMR8104, INSERM U1016, Université de Paris, 75019 Paris, France; 5Équipe Labellisée Ligue Nationale Contre le Cancer (LNCC), 75014 Paris, France

**Keywords:** COVID-19, hematology, immunotherapy

## Abstract

Background: Patients with hematological malignancies are at greater risk of severe COVID-19 and have been prioritized for COVID-19 vaccination. A significant proportion of them have an impaired vaccine response, both due to the underlying disease and to the treatments. Methods: We conducted a prospective observational study to identify the specific risks of the outpatient population with hematological diseases. Result: Between 22 December 2021 to 12 February 2022, we followed 338 patients of which 16.9% (*n* = 57) developed SARS-CoV-2 infection despite previous vaccination (94.7%). COVID-19 patients were more likely to have received immunotherapy (85.5% vs. 41%, *p* < 10^−4^), and particularly anti-CD20 monoclonal antibodies (40% vs. 14.9%, *p* < 10^−4^) and Bruton’s tyrosine kinase inhibitors (BTKi) (7.3% vs. 0.7%, *p* < 10^−2^). There was no significant difference in demographic characteristics or hematological malignancies between COVID-19-positive and non-positive patients. Patients hospitalized for COVID-19 had more frequently received immunotherapy than patients with asymptomatic or benign forms (100% vs. 77.3%, *p* < 0.05). Hospitalized COVID-19 patients had a higher proportion of negative or weakly positive serologies than non-hospitalized patients (92.3% vs. 61%, *p* < 0.05). Patients who received tixagevimab/cilgavimab prophylaxis (*n* = 102) were less likely to be COVID-19-positive (4.9 vs. 22%, *p* < 0.05) without significant difference in hospitalization rates. Conclusion: In the immunocompromised population of patients with hematological malignancies, the underlying treatment of blood cancer by immunotherapy appears to be a risk factor for SARS-CoV-2 infection and for developing a severe form.

## 1. Introduction

The risk of severe forms of COVID-19 and death is higher among patients treated for hematological disease than in the general population [1,2,3,4,5,6]. COVID-19 vaccination campaigns helped contain the pandemic by reducing hospitalizations and deaths, but immunocompromised populations remain particularly exposed due to altered vaccine responses [7,8,9,10]. A high proportion of patients with hematological malignancies respond poorly to vaccination due to the underlying disease and its treatment [11,12,13]. Immunotherapy, such as anti-CD38 and anti-CD20 monoclonal antibodies (mAb), are associated with a low rate of neutralizing antibodies (nAb) after two or three doses of vaccination [14]. In addition, SARS-CoV-2 variants demonstrated partial vaccine escape, especially the Omicron (BA.1) variant as compared to the delta (B.1.617.2) variant [15]. This variant also resisted anti-SARS-CoV-2 mAb prophylaxis [16]. Omicron was designated a variant of concern by WHO on 26 November 2021 and is now the most prevalent lineage globally [17]. Data suggest a decreased severity of COVID-19 disease, with a decoupling of cases and death, compared to previous waves [18]. However, the risk of severe forms of SARS-CoV-2 Omicron variant infection in a population of immunocompromised hematology patients is not known. In the context of the fast spread of new Omicron variants, we conducted a specific observation of COVID-19 outcomes in our hematological department.

## 2. Materials and Methods

### 2.1. Study Population

We conducted a prospective observational study in the hematological outpatient clinic of Cochin Hospital (University Hospitals of Paris, France) from 22 December 2021 to 13 February 2022. The study protocol was approved by the institutional review board of Cochin Hospital (IRB-CLEP N: AAA-2022-08033). Our study population consisted of all adults (≥18 years) of the hematological outpatient clinic with informed consent to participate. In routine care since the COVID-19 epidemic, they were systematically screened for COVID-19 infection before being admitted to the outpatient clinic. At home, they were trained to be screened for infection by COVID-19 in case of close contact with a positive case or if they presented even mild symptoms. They informed the hematologists by a dedicated telephone line if test is positive. During the 7-week study period, the Omicron variant was predominant (>99%) in the Ile-de-France region [19]. The peak incidence rate occurred around 9 January 2022 in the region [19]. At the end of the study, the cumulative incidence of Omicron-positive cases was estimated to be 15.8% in the French population [20].

### 2.2. Immunoprophylaxis Procedures

In France, immunocompromised patients with hematological diseases were given priority to receive anti-SARS-CoV-2 vaccination from January 2021. Starting September 2021, they were advised to receive all three doses. Moreover, anti-SARS-CoV-2 mAbs became potential options for COVID-19 immunoprophylaxis from August 2021. The casirivimab/imdevimab combination was first used for pre-exposure or post-exposure prophylaxis or for the treatment of mild to moderate disease [21] but lacked neutralization against new dominant Omicron variant [16]. On 9 December 2021, tixagevimab/cilgavimab received early access authorization for preexposure prophylaxis, then for active infection because of a maintained sensitivity of Omicron (BA.1 and BA.2) variants [16,22]. The tixagevimab/cilgavimab authorization was updated in March 2022 with the emergence of the Omicron BA.4 and BA.5 variants not sufficiently neutralized by Evusheld, requiring the dosages to be doubled [23]. These treatments were reserved for patients at risk of severe forms and for poor responders to vaccination i.e., having an antibody level <260 BAU after three or four vaccine doses.

### 2.3. Data Collection and Outcomes

We recorded age, sex, body mass index (BMI), hematological malignancy, hematological treatment, lymphocytes and gammaglobulins count, SARS-CoV-2 vaccination status, anti-SARS-CoV-2 spike immunoglobulin G (IgG) level, and date of positive SARS-CoV-2 polymerase chain reaction (PCR) test. We defined SARS-CoV-2 infection as testing positive >14 days after the completion of the recommended vaccine series. For each COVID-19-positive patient, we recorded the following data attributable to COVID-19: hospital admissions, intensive care unit (ICU) admissions, mechanical ventilation, and mortality.

The primary outcome of the study was the SARS-CoV-2 infection rate. Secondary outcomes were hospitalization and mortality rates for COVID-19. Outcomes were analyzed on the total cohort and on the stratified sample of patients who received immunoprophylaxis.

### 2.4. Statistical Analyses

Patient’s characteristics and outcomes were described with the proportions for categorical variables and with the means, medians, standard deviation (SD), and intervalley quartile ratio (IQR) for quantitative variables. Clinical characteristics were compared between different groups using a z-test. Furthermore, *p*-values less than 0.05 were considered statistically significant. Analyses were performed with Addinsoft v.2020 (XLSTAT, New York, NY, USA).

## 3. Results

### 3.1. Patient Population and COVID-19-Positive Cases

Between 22 December 2021 and 12 February 2022, we identified 338 patients followed in our day-hospital unit, referred to as the outpatient population and including 40% for multiple myeloma (*n* = 135), 27% for myelodysplastic syndrome and acute leukemia (*n* = 92), 20% for indolent lymphoma and chronic lymphocytic leukemia (*n* = 67), 11% for Hodgkin’s lymphoma and high-grade non-Hodgkin’s lymphoma (*n* = 39). Among them, 80% required treatment for a blood disease and 15% benefited from a blood transfusion. Treatments in progress or received in the last 12 months were chemotherapy alone (*n* = 182, 52%), T-cell or B-cell immunotherapy (*n* = 162, 46.3%) including anti-CD38 drugs (*n* = 83, 23.7%), anti-CD20 drugs (*n* = 66, 18.9%), and other immunotherapies (e.g., brentuximab, nivolumab, blinatunumab, *n* = 13, 3.7%).

During this 7-week period, 57 patients (16,9%) developed SARS-CoV-2 infection. The median age of COVID-19-positive patients was 71 years (interquartile range [IQR], 63–78 years), with 21 (36.8%) patients being 75 years of age or older (Table 1) and 36 (63%) being men.

Demographic characteristics of COVID-19-positive patients were representative of the outpatient population, with no significative differences concerning age or gender. The proportion of COVID-19 patients treated with immunotherapy in the past year was higher compared to the non-positive COVID-19 outpatient population (85.5% vs. 41%, *p* < 10^−4^) (Figure 1). In particular, COVID-19 patients were more likely to have received anti-CD20 mAbs (40% vs. 14.9%, *p* < 10^−4^) and Bruton’s tyrosine kinase inhibitors (BTKi) (7.3% vs. 0.7%, *p* < 10^−2^).

There was no difference in the distribution of hemopathies between COVID-19 patients and the general outpatient population, although we observed a trend toward a higher proportion of multiple myeloma (46% vs. 39%, *p* = 0.34) or chronic indolent B cell lymphoma (26% vs. 19%, *p* = 0.18) among COVID-19 patients (Figure 2).

### 3.2. Hospitalization and Mortality Due to COVID-19

Among the 57 patients diagnosed with SARS-CoV-2 infection, 22.8% (*n* = 13) were hospitalized for a severe form of COVID-19, of which 84.6% (*n* = 11) required invasive mechanical ventilation. Among them, 23% (*n* = 3) died. Table 1 summarizes the differences between patients with a COVID-19 asymptomatic or mild illness and those with a severe disease requiring hospitalization. Repartition of haematological malignancy was not statistically different between hospitalized and non-hospitalized patients. However, patients receiving T-cell or B-cell immunotherapy accounted for the totality of hospitalization cases (*n* = 13, of which 6 were treated by anti-CD20, 5 by anti-CD38, 1 by anti-CD19/anti-CD3, and 1 by anti CD52 mAb) and were found more frequently in this group than in the group of patients with asymptomatic or benign forms (100% vs. 77.3%, *p* < 0.05).

All three COVID-19-positive patients who died were men aged from 76 to 83 years old, and all had negative anti-SARS-CoV-2 serology despite a complete vaccination schedule. They all received immunotherapy: anti-CD38 mAb for multiple myeloma (daratumumab, *n* = 1), anti-CD52 mAb for T-cell chronic leukemia (alemtuzumab, *n* = 1), and anti-CD19 anti-CD3 bispecific mAb for acute lymphoblastic leukemia (blinatumomab, *n* = 1).

The management of COVID-19 involved specific anti-SARS-CoV-2 treatments, the monoclonal antibodies tixagevimab/cilgavimab (*n* = 5) and sotrovimab (*n* = 4) and the antiviral nirmatrelvir/ritonavir (*n* = 3). These treatments have been proposed for patients meeting the eligibility criteria for early-access procedures, in particular early symptomatic forms, whether severe or mild. They constituted 23.1% of hospitalized patients and 20.5% of non-hospitalized patients (Table 1). One patient died despite curative treatment with tixagevimab/cilgavimab.

### 3.3. Effects of Anti-SARS-CoV-2 Serology and mAb Prophylaxis

A large majority of COVID-19-positive patients (*n* = 54, 94.7%) had been vaccinated against SARS-CoV-2, including 87% (*n* = 47) with full vaccination, i.e., having received the booster dose after three months of the initial scheme. Most received a complete BNT162b2 (Pfizer-BioNTech) vaccine regimen (*n* = 42); nine received a mixed BNT162b2/mRNA-1273 (Moderna) regimen; two received the ChAdOx1-S (AstraZeneca) vaccine, and one received the Ad26COV2.S (Janssen) vaccine. Three patients did not receive any vaccination dose, and ten patients received only two doses. All patients (13/13) requiring hospitalization secondary to COVID-19 illness received full vaccination with three or four doses.

Hospitalized COVID-19 patients had a higher proportion of negative or weakly positive serologies (*n* = 12) than the group of non-hospitalized patients (*n* = 27) (92.3% vs. 61%, *p* < 0.05). Median time between last vaccination and COVID-19 infection was similar in hospitalized and non-hospitalized patients.

During the study period, we treated 102 patients prophylactically with tixagevimab/cilgavimab among the 338 patients in our outpatient population. A greater proportion of patients who did not receive prophylactic treatment tested positive (22% vs. 4.9%, *p* < 0.05). However, the proportion of positive patients who were hospitalized was not different between those who received prophylaxis or not. Indeed, among the patients who received mAb prophylaxis, one was hospitalized with a need for invasive mechanical ventilation support and one died.

## 4. Discussion

Real-world data on SARS-CoV-2 Omicron impact in patients with hematological malignancies are limited [24]. Our single-center observational study suggests that the Omicron variant may be responsible for severe forms of COVID-19. In this frail population, 23% of COVID-19-positive patients required hospitalization, and 3/57 patients died despite epidemiological data suggesting that the illness may be milder than the previous variant in the general population [25]. Patients with hematological disease were given priority to receive anti-SARS-CoV-2 vaccination from January 2021, with high adhesion to booster reinjection: 95% and 87% had a complete vaccination status and a booster dose, respectively. They usually maintained masking and social distancing in their daily lives. The cumulative incidence of COVID-19 cases in the Omicron era was 16.9% in our frail population, compared to a rate estimated at 15.8% in the general population [20]. Mortality was lower than rates reported in vaccinated adult patients with hematological malignancies, explained by the study periods used in this population during the predominant and more lethal alpha-variant era [26]. A limitation of our study was the small sample size of COVID-19-positive hematological patients.

In our study, the main treatment difference between groups with severe or mild forms of COVID-19 was T-cell or B-cell immunotherapy directed against plasma cell or B/T lymphocytes. Underlying hematological malignancies or usual risk factors, such as age, sex, and lymphocyte count, were not different between the hospitalized and non-hospitalized groups. The three patients who died had no detectable humoral response to vaccination. In vitro data about nAb against the Omicron variant were consistent with our experience: in only 56% of patients with hematological malignancies, nAb titers against Omicron are detectable after booster vaccination [27]. Moreover, treatment seems to be an important predictor of vaccination response. A multivariable logistic regression analysis in patients with blood cancer showed treatment with anti-CD20 mAbs within 12 months and BTKi within 28 days of the third vaccine dose were significantly associated with undetectable nAb against Omicron, while hematological malignancies and age were not [27]. A limitation of our study is the lack of adjustment for other factors besides treatments that are known to explain the risk of a patient developing severe disease following diagnosis with COVID-19.

COVID-19 infection still represents a challenge for caring for immunocompromised patient. Clinical presentation is different from other respiratory viruses with a higher hospitalization rate and mortality [28]. Prophylaxis strategies are warranted to protect immunocompromised patients. Social distancing is effective but cannot be rigorously adhered to in the context of chronic disease and the risk of further waves of the epidemic. Monoclonal antibodies are potential options for COVID-19 immunoprophylaxis. However, clinical trials included few immunocompromised patients, and the alpha variant was predominant [22]. At the time of our study, approximatively 30% benefited from tixagevimab/cilgavimab, which remained effective against Omicron [16]. This immunoprophylaxis campaign appears to be effective, given that patients who received this mAbs combination were significantly less infected than those who did not.

## 5. Conclusions

During the Omicron era, special attention must be paid to immunocompromised patients who remain at risk. These risks are increased by underlying diseases that limit the effectiveness of vaccines but also, as our study shows, by current or past treatments and, in particular, by immunotherapies. The knowledge and identification of these risks are essential in order to inform patients and propose the best pre-emptive strategies, such as social distancing, vaccination, and/or monoclonal antibodies prophylaxis.

## Figures and Tables

**Figure 1 viruses-14-02377-f001:**
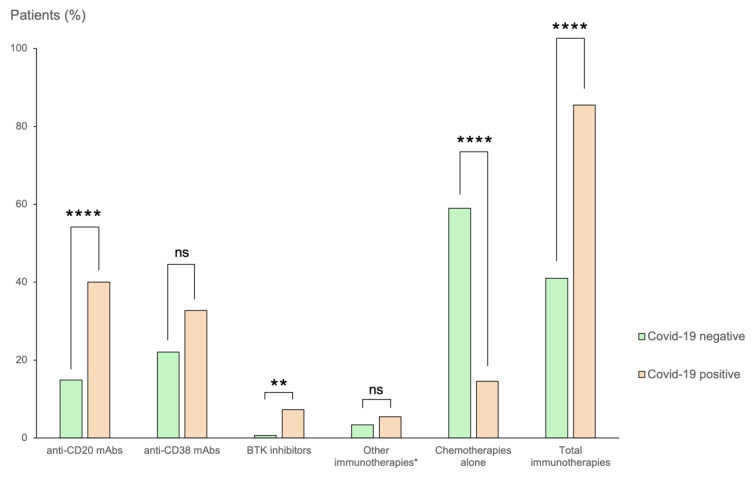
Distribution of hematological treatments (%) in COVID-19-positive and non-positive patients. Abbreviations: mAbs, monoclonal antibodies; BTK, Bruton’s tyrosine kinase. (**, *p* < 10^−2^; ****, *p* < 10^−4^; ns, non-significant.)

**Figure 2 viruses-14-02377-f002:**
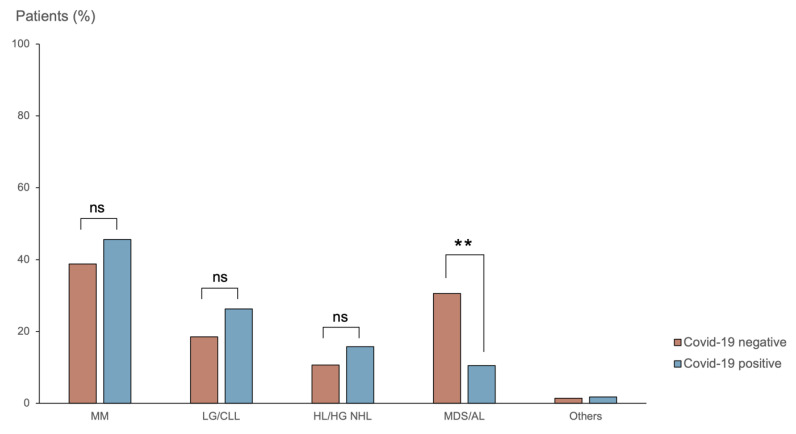
Distribution of hematological malignancies (%) in COVID-19-positive and non-positive patients. Abbreviations: MM, myeloma multiple; LG/CLL, low-grade lymphoma and chronic lymphocytic leukemia; HL/HG NHL, Hodgkin’s lymphoma and high-grade non-Hodgkin’s lymphoma; MDS/AL, myelodysplastic syndrome and acute leukemia. (**, *p* < 10^−2^; ns, non-significant).

**Table 1 viruses-14-02377-t001:** COVID-19-positive patients characteristics.

	Outpatients	Inpatients
	*n* = 44	*n* = 13
Median age (IQR)—yr	69 (60.5–78)	75 (69–77)
Age category—no. (%)		
<75 ans	30 (68)	6 (46.2)
> or = 75 ans	14 (31.8)	7 (53.8)
Male sex—no. (%)	24 (54.5)	12 (92.3)
Median BMI (IQR)	24.1 (22.5–26.3)	23.5 (22.5–26.3)
Hematologic malignancy—no. (%)		
Multiple myeloma	20 (45.5)	5 (38.5)
Low-grade lymphoma	7 (15.9)	3 (23.1)
High-grade lymphoma	8 (18.2)	1 (7.7)
Chronic lymphocytic leukemia	3 (6.8)	2 (15.4)
Others *	6 (13.6)	2 (15.4)
On treatment—no. (%)		
Anti-CD20	16 (36.4)	6 (46.2)
Anti-CD38	15 (34.1)	3 (23.1)
IBTK	2 (4.5)	2 (15.4)
Other immunotherapies **	1 (2.3)	2 (15.4)
Others ***	8 (18.2)	-
Median lymphocyte count (IQR)—G/L	0.79 (0.47–1.03)	0.5 (0.4–1.1)
Missing information—no (%)	11 (25)	1 (7.7)
Hypogammaglobulinemia—no. (%)		
Present	19 (43.2)	8 (61.5)
Absent	7 (15.9)	1 (7.7)
Missing information	18 (40.9)	4 (30.1)
Vaccination—no. (%)		
Initial scheme	41	13
Full scheme	34	13
Serology—no. (%)		
Negative	22 (50)	9 (69)
<264 BAU	4 (9.1)	2 (15.4)
264—1000 BAU	1 (2.3)	1(7.7)
>1000 BAU	6 (13.6)	1(7.7)
Missing information	11 (25)	-
Casirivimab/imdevimab prophylaxis—no. (%)	5 (11.4)	2 (15.4)
Tixagevimab/cilgavimab prophylaxis—no. (%)	3 (6.8)	2 (15.4)
Median duration between last vaccination and COVID-19 diagnosis (IQR)—days	117 (66–151)	84 (49–107)
Anti-SARS-CoV-2 curative specific treatment—no. (%)	9 (20.5)	3 (23.1)
Tixagevimab/cilgavimab	3 (6.8)	2 (15.4)
Sotrovimab	3 (6.8)	1 (7.7)
Nirmatrelvir/ritonavir	3 (6.8)	-
Outcomes		
Invasive mechanical ventilation	-	11 (84.6)
Deaths	-	3 (23)

* Acute lymphoid leukemia (*n* = 3), chronic myelomonocytic leukemia (*n* = 1), Evans’ syndrome (*n* = 1), myelodysplastic syndrome (*n* = 3). ** Alemtuzumab, blinatumomab, elotuzumab. *** Chemotherapy only, ciclosporine, steroids. Abbreviations: BMI, body mass index; BAU, binding antibody units; IBTK, Inhibitor of Bruton tyrosine kinase; IQR, interquartile range; no, number; yr, years.

## Data Availability

Data are available upon reasonable request.

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
