# Peer review of "Patients with Hematological Malignancies Treated with T-Cell or B-Cell Immunotherapy Remain at High Risk of Severe Forms of COVID-19 in the Omicron Era"

_viruses, 2022, doi:10.3390/v14112377_

Round 1
Reviewer 1 Report
The authors report on a small cohort of patients who developed COVID-19 between December 2021 and February 2022. The majority of infections occurring at this time would have been of the Omicron form and so results continue to be relevant to the current phase of the pandemic.
Given the very different findings for checkpoint inhibitor immunotherapy used to treat solid cancers I would suggest the authors discuss anti CD20 and anti-CD 38 as immunomodulatory drugs to ensure proper distinction of these drugs and to be consistent with previously published data on the impact of treatment of haematological cancers on outcomes.
I suggest the title is changed to:
Patients with haematological malignancies treated with immunomodulatory drugs remain at high risk of severe forms of Covid-19 in the omicron era.
I would suggest that patients on anti CD20 or anti CD38 be compared to the rest of the study population having removed those on other anti-cancer treatments such as nivolumab. Consider investigating the two drugs separately or explain why you could not/did not.
Authors should reference: Breakthrough COVID-19 in vaccinated patients with hematologic malignancies: results from EPICOVIDEHA survey, Pagano et al and explain reasons why their results are different to this paper.
I don’t think the authors have made any adjustment for other factors that influence outcome such as age, sex and haematological cancer type. It is not sufficient to say these are similar in the two groups, particularly given the small sample size involved. This is a major limitation which should be addressed by repeating the analysis using regression models where covariates can be included.
The authors also need to tone down this statement in the discussion “Our single-centre observational study suggest that omicron 184 variant may be responsible for high mortality and morbidity”. 3 deaths in 57 covid positive haematological cancer patients is a mortality rate of 5%. This is much lower than seen in haematological patients in earlier waves of the pandemic when beta and delta strains predominated and the authors should point this out referencing: https://pubmed.ncbi.nlm.nih.gov/35188551/ https://pubmed.ncbi.nlm.nih.gov/32853557/ https://pubmed.ncbi.nlm.nih.gov/32737082/
https://pubmed.ncbi.nlm.nih.gov/33113551/
Vaccination and less severe coronavirus strains has left haematological cancer patients as a group better protected. The authors are of course right to point out that some patients do not generate an antibody response, in some cases because of recent treatment.
The authors also need to add to the discussion that their small samples size of COVID positive haematological patients is a big limitation to this study.

Author Response
The authors report on a small cohort of patients who developed COVID-19 between December 2021 and February 2022. The majority of infections occurring at this time would have been of the Omicron form and so results continue to be relevant to the current phase of the pandemic.
1. Given the very different findings for checkpoint inhibitor immunotherapy used to treat solid cancers I would suggest the authors discuss anti CD20 and anti-CD 38 as immunomodulatory drugs to ensure proper distinction of these drugs and to be consistent with previously published data on the impact of treatment of haematological cancers on outcomes. I suggest the title is changed to: Patients with haematological malignancies treated with immunomodulatory drugs remain at high risk of severe forms of Covid-19 in the omicron era.
Response from the authors: We totally agree with the reviewer. None of our patient with covid-19 infection were treated with checkpoint inhibitor. As the reviewer emphasize, immunotherapy that have been used for our patients are indeed immunosuppressive with anti-plasma or anti-lymphocytes properties. The title of the revised manuscript was replaced by “Patients with haematological malignancies treated with immunosuppressive immunotherapy remain at high risk of severe forms of Covid-19 in the omicron era.” rather than immunomodulatory drugs which is the class of "IMiDs" in hematology (lenalidomide, pomalidomide, thalidomide).
2. I would suggest that patients on anti CD20 or anti CD38 be compared to the rest of the study population having removed those on other anti-cancer treatments such as nivolumab. Consider investigating the two drugs separately or explain why you could not/did not.
Response from the authors: We thank the reviewer to raise this point. In fact, the anti-CD20 and anti-CD38 monoclonal antibodies were the most used in the study patients but were not the only ones (also anti-CD3/CD19 and anti-CD52) and for this reason we cannot not isolate them to compare them to other immunotherapies. We have clarified for the reader by adding the following sentence “However, patients receiving immunosuppressive immunotherapy accounted for the totality of hospitalization cases (n = 13, of which 6 treated by anti-CD20, 5 by anti-CD38, 1 by anti-CD19/anti-CD3, 1 by anti CD52 mAb” in the revised manuscript.
3. Authors should reference: Breakthrough COVID-19 in vaccinated patients with hematologic malignancies: results from EPICOVIDEHA survey, Pagano et al and explain reasons why their results are different to this paper.
Response from the authors: We thank the reviewer to detect this reference. The Pagano et al paper was added as reference in the revised version, and the following sentence was added : “Mortality was lower than rates reported in vaccinated adult patients with hematological malignancies, explained by study periods conducted in this population during the predominant and more lethal alpha variant era [25].”
4. I don’t think the authors have made any adjustment for other factors that influence outcome such as age, sex and haematological cancer type. It is not sufficient to say these are similar in the two groups, particularly given the small sample size involved. This is a major limitation which should be addressed by repeating the analysis using regression models where covariates can be included.
Response from the authors: We thank the reviewer to raise this point. We did not use a logistic regression model, because of small sample size. As the reviewer reminds, the study did not allow to identify the factors characterizing a group of subjects at higher risk. In fact, the literature has already reported these factors. Our study objective was to focus on the comparison between cancer treatments in order to identify if some had a greater impact on the Covid-19 outcomes. The following sentences were added in the revised manuscript : “In our study, the main treatment difference between groups with severe or mild forms of Covid-19 was immunosuppressive immunotherapy. Underlying haematological malignancies or usual risk factors such as age, sex or lymphocytes count were not different between the hospitalized and non-hospitalized groups”, line 199, and “A limitation of our study is the absence of a regression model to identify factors other than treatments, characterizing patients with severe forms”, line 210.
5. The authors also need to tone down this statement in the discussion “Our single-centre observational study suggest that omicron 184 variant may be responsible for high mortality and morbidity”. 3 deaths in 57 covid positive haematological cancer patients is a mortality rate of 5%. This is much lower than seen in haematological patients in earlier waves of the pandemic when beta and delta strains predominated and the authors should point this out referencing: https://pubmed.ncbi.nlm.nih.gov/35188551/ https://pubmed.ncbi.nlm.nih.gov/32853557/ https://pubmed.ncbi.nlm.nih.gov/32737082/ https://pubmed.ncbi.nlm.nih.gov/33113551/
Response from the authors: We totally agree with the reviewer. The statement “may be responsible for high mortality and morbidity” was deleted in the revised version and replaced by “Our single-centre observational study suggest that omicron variant may be responsible for severe forms of Covid-19”. We agree with the lower rates of mortality and the previous correction incorporated this observation “Mortality was lower than rates reported in vaccinated adult patients with hematological malignancies, explained by study periods conducted in this population during the predominant and more lethal alpha variant era [25].” We have not referenced mortality rates from earlier waves studied in the pre-vaccination era.
6. Vaccination and less severe coronavirus strains has left haematological cancer patients as a group better protected. The authors are of course right to point out that some patients do not generate an antibody response, in some cases because of recent treatment.
Response from the authors: We thank the reviewer for this observation.
7. The authors also need to add to the discussion that their small samples size of COVID positive haematological patients is a big limitation to this study.
Response from the authors: We thank the reviewer to detect this missing. The following sentence was added in the revised manuscript: “A limitation of our study was the small samples size of Covid-19 positive haematological patients”, line 197.
Reviewer 2 Report
The article is overall well written. The subject is really interesting, indeed few studies had focused on the specific population of hematological diseases patients. The results are expected but deserve to be published.
I would mainly suggest working on the discussion.
The first sentence:" Real-world data on SARS-CoV-2 omicron impact in patients with hematological malignancies are limited" is difficult to understand. What do you mean by Real-world?
It lacks focus on the evolution of prophylaxis strategies according to omicron sub-lineages resistance.
It also would be interesting to compare the mortality of hematological patients treated with immunotherapy and infected by other respiratory viruses. Just a few sentences, no need to expand on the subject.
Author Response
The article is overall well written. The subject is really interesting, indeed few studies had focused on the specific population of hematological diseases patients. The results are expected but deserve to be published.
1. I would mainly suggest working on the discussion.
Response from the authors: We thank the reviewer to raise this point. The following sentences was added in the discussion part : “Mortality was lower than rates reported in vaccinated adult patients with haematological malignancies, explained by study periods conducted in this population during the predominant and more lethal alpha variant era [26]. A limitation of our study was the small samples size of Covid-19 positive haematological patients.”, line 197 ; “In our study, the main treatment difference between groups with severe or mild forms of Covid-19 was immunotherapy directed against plasma cell or B/T lymphocytes . Underlying haematological malignancies or usual risk factors such as age, sex or lymphocytes count were not different between the hospitalized and non-hospitalized groups.”, line 202 ; and “A limitation of our study is the absence of a regression model to identify factors other than treatments, characterizing patients with severe forms.” line 213.
2. The first sentence:" Real-world data on SARS-CoV-2 omicron impact in patients with hematological malignancies are limited" is difficult to understand. What do you mean by Real-world?
Response from the authors: We thank the reviewer for this observation. The term "real-world" refers to an expression commonly used in the medical literature to designate a study in usual care, i.e. outside of a clinical trial.
3. It lacks focus on the evolution of prophylaxis strategies according to omicron sub-lineages resistance.
Response from the authors: We thank the reviewer to detect this missing. The following sentence was added in the revised manuscript “The tixagevimab/cilgavimab authorization is updated in March 2022 with the emergence of the omicron BA.4 and BA.5 variants not sufficiently neutralized by evusheld, requiring the dosages to be doubled [23].”, line 77. Reference 23. Food and Drug Administration. FDA releases important information about risk of COVID-19 due to certain variants not neutralized by Evusheld. Available online: https://www.fda.gov/drugs/drug-safety-and-availability/fda-releases-important-information-about-risk-covid-19-due-certain-variants-not-neutralized-evusheld (accessed on October 13 2022).
4. It also would be interesting to compare the mortality of hematological patients treated with immunotherapy and infected by other respiratory viruses. Just a few sentences, no need to expand on the subject.
Response from the authors: We thank the reviewer for this observation. At the start of the study, we did not identify infection by other respiratory viruses as an outcome to be collected. We cannot find these results retrospectively. However literature data comparing Covid-19 and seasonal influenza show longer hospitalization duration, need for mechanical ventilation and death. This difference is even more important for immunocompromised patient (Comparison of clinical characteristics and disease outcome of COVID-19 and seasonal influenza, Brehm Scientific reports 2021). A sentence was added in the discussion line 214-215 : « Clinical presentation, is different from other respiratory viruses with longer hospitalization rate and mortality » (Piroth, The Lancet.respiratory Medecine, 2021, Comparison of the characteristics, morbidity, and mortality of COVID-19 and seasonal influenza: a nationwide, population-based retrospective cohort study), highlighting the need for special prevention in immunocompromised patients.
Round 2
Reviewer 1 Report
I am generally happy with the authors response to my comments. I have two things that I would still suggest be addressed:
1. Title. I appologise for my mistake suggesting immunomodulatory to describe the anti T-cell and anti- B cell therapies that the authors are analysing. The authors are quite right this is used to describe different drugs. Whilst their suggestion of immunosupressive immunotherapy is fine I cant really see that this is a commonly used term to describe anti T cell and anti-B cell drugs. I suggest the title be updated further to: Patients with haematological malignancies treated with T-cell depleting or B-cell depleting therapies remain at high risk of severe forms of Covid-19 in the omicron era.
The authors should use T-cell depleting or B-cell depleting therapies to describe these drugs in the main text as well.
Regarding my 4th comment (4. I don’t think the authors have made any adjustment for other factors that influence outcome such as age, sex and haematological cancer type. It is not sufficient to say these are similar in the two groups, particularly given the small sample size involved. This is a major limitation which should be addressed by repeating the analysis using regression models where covariates can be included.
I was not concerned about the authors not identifying new risk factors, rather I was concerned that known risk factors had not been adjusted for by including them as covariates in a logistic regression model. I would like the authors to change
“A limitation of our study is the absence of a regression model to identify factors other than treatments, characterizing patients with severe forms”, line 210.
to "A limitation of our study is the lack of adjustment for other factors besides treatments that are known to explain risk of a patient developing severe disease following diagnosis with COVID-19."
Author Response
I am generally happy with the authors response to my comments. I have two things that I would still suggest be addressed:
1. Title. I appologise for my mistake suggesting immunomodulatory to describe the anti T-cell and anti- B cell therapies that the authors are analysing. The authors are quite right this is used to describe different drugs. Whilst their suggestion of immunosupressive immunotherapy is fine I cant really see that this is a commonly used term to describe anti T cell and anti-B cell drugs. I suggest the title be updated further to: Patients with haematological malignancies treated with T-cell depleting or B-cell depleting therapies remain at high risk of severe forms of Covid-19 in the omicron era. The authors should use T-cell depleting or B-cell depleting therapies to describe these drugs in the main text as well.
Response from the authors: We agree with the reviewer. However, the term "depleting" is not always appropriate to the mechanisms of action of the drugs concerned in the study. The title of the revised manuscript was replaced by “Patients with haematological malignancies treated with T-cell or B-cell immunotherapy remain at high risk of severe forms of Covid-19 in the omicron era.” The terms "T-cell or B-cell" were added in the revised manuscript, line 110, line 150 and line 203.
2. Regarding my 4th comment (4. I don’t think the authors have made any adjustment for other factors that influence outcome such as age, sex and haematological cancer type. It is not sufficient to say these are similar in the two groups, particularly given the small sample size involved. This is a major limitation which should be addressed by repeating the analysis using regression models where covariates can be included. I was not concerned about the authors not identifying new risk factors, rather I was concerned that known risk factors had not been adjusted for by including them as covariates in a logistic regression model. I would like the authors to change “A limitation of our study is the absence of a regression model to identify factors other than treatments, characterizing patients with severe forms”, line 210. to "A limitation of our study is the lack of adjustment for other factors besides treatments that are known to explain risk of a patient developing severe disease following diagnosis with COVID-19."
Response from the authors: We thank the reviewer for the suggestion. The sentence was changed as suggested by the reviewer in the revised manuscript "A limitation of our study is the lack of adjustment for other factors besides treatments that are known to explain risk of a patient developing severe disease following diagnosis with COVID-19.", line 213.